# The Role of Diet in the Pathogenesis and Management of Inflammatory Bowel Disease: A Review

**DOI:** 10.3390/nu13010135

**Published:** 2020-12-31

**Authors:** Gabrielle Wark, Dorit Samocha-Bonet, Simon Ghaly, Mark Danta

**Affiliations:** 1St Vincent’s Clinical School, UNSW, Sydney, NSW 2052, Australia; Gabrielle.Wark@svha.org.au (G.W.); d.samochabonet@garvan.org.au (D.S.-B.); Simon.Ghaly@svha.org.au (S.G.); 2Department of Gastroenterology and Hepatology, St Vincent’s Hospital, Sydney, SW 2010, Australia; 3Garvan Institute of Medical Research, Sydney, NSW 2010, Australia

**Keywords:** inflammatory bowel disease (IBD), Crohn’s disease (CD), ulcerative colitis (UC), diet, macronutrients, gut microbiome, gut epithelium, gut immunity

## Abstract

Inflammatory bowel diseases, which include ulcerative colitis and Crohn’s disease, are chronic relapsing and remitting inflammatory diseases of the gastrointestinal tract that are increasing in prevalence and incidence globally. They are associated with significant morbidity, reduced quality of life to individual sufferers and are an increasing burden on society through direct and indirect costs. Current treatment strategies rely on immunosuppression, which, while effective, is associated with adverse events. Epidemiological evidence suggests that diet impacts the risk of developing IBD and modulates disease activity. Using diet as a therapeutic option is attractive to patients and clinicians alike due to its availability, low cost and few side effects. Diet may influence IBD risk and disease behaviour through several mechanisms. Firstly, some components of the diet influence microbiota structure and function with downstream effects on immune activity. Secondly, dietary components act to alter the structure and permeability of the mucosal barrier, and lastly dietary elements may have direct interactions with components of the immune response. This review will summarise the mechanisms of diet–microbial–immune system interaction, outline key studies examining associations between diet and IBD and evidence demonstrating the impact of diet on disease control. Finally, this review will outline current prescribed dietary therapies for active CD.

## 1. Background

The inflammatory bowel diseases (IBD) ulcerative colitis (UC) and Crohn’s disease (CD) are characterised by chronic relapsing and remitting inflammation of the gastrointestinal tract. UC is limited to the mucosal layer of the colon, invariably involving the rectum and may extend to involve more proximal portions of the bowel in a continuous fashion. CD is characterised by transmural inflammation and skip areas of involvement (segments of normal-appearing bowel interrupted by areas of disease) of the entire gastrointestinal tract from the mouth to the anus [1,2]

Patients suffering from these conditions present with symptoms such as abdominal pain, bloody diarrhoea and weight loss. The symptoms are associated with significant morbidity including intestinal perforation, strictures, fistulising disease and long-term risk of malignancy. In the paediatric setting, active disease has the ability to impact upon growth potential as well as pubertal and emotional development [3]. As a result of the gastrointestinal tract involvement, malnutrition is a common issue in patients with IBD and may occur as a result of reduced oral intake, accelerated gastrointestinal losses and increased nutritional requirements during active disease [4]. Colonoscopy including mucosal biopsy remains the gold standard for diagnosis and assessment of disease activity. However, non-invasive biomarkers of inflammation in the form of serum C-reactive protein and faecal calprotectin can also be helpful tools in monitoring disease activity [5].

The aetiology of IBD is a complex interaction of multiple factors including genetics, immune system, microbial and environmental factors, in particular diet. At a genetic level, mutations in one or more of over 200 genes coding for or modulating protein expression have been identified as impacting the immune regulatory functions in individuals with IBD [6]. When exposed to an environmental trigger, this may lead to a dysregulated response by the mucosal immune system to the microbiota that reside within the intestinal lumen, culminating in an inflammatory response [2,6,7]. There is intense interest expressed by both patients and clinicians in understanding the role that diet may play in the pathogenesis and management of IBD, which may be translated into improving outcomes in individuals with IBD [7,8].

Twin studies demonstrate modest concordance rates in IBD even among monozygotic twins, with the rates higher in CD (20–55%) compared to UC (6.3–17%) [9,10]. This highlights the limited impact of genetics and the potential importance of environmental factors in the pathogenesis of IBD. Epidemiological studies reveal a higher prevalence and incidence of IBD in developed nations. Incidence rates, however, are accelerating in more recently Westernised countries such as Brazil, Taiwan and urban Korea and the ratio of diagnosing UC to CD is decreasing, mirroring the same pattern that evolved in the 20th century in the Western world [11,12,13]. There is also a higher risk in first-generation immigrants from developing countries who have relocated to developed nations, suggesting the typical Western lifestyle including diet and other environmental factors may contribute to the development of IBD [14,15]. This is supported in the literature, where it has been shown that exposures such as use of antibiotics during childhood, vitamin D level and latitude of residence, smoking and oral contraceptive use have all been associated with an increased risk of developing IBD [16,17,18].

## 2. Treatment of IBD

Current standard of care in IBD is determined by the subtype, severity, location of gastrointestinal tract involvement and complications of the disease. The medical management includes anti-inflammatory, immune modulatory and biologic therapies. These are directed against several targets. First-line therapy in UC is oral and rectally delivered aminosalicylate therapy, which is an intestinal anti-inflammatory agent. If ineffective, and in patients with an underlying diagnosis of CD, immune modulator therapy is recommended. These include anti-metabolites (Thiopurines and Methotrexate), anti-tumour necrosis factor (TNF-α, e.g., Infliximab, Adalimumab), α_4_β_7_ ligand blocker (Vedolizumab), Interleukin (IL)-12 and IL-23 blocker (Ustekinumab) and Janus Kinase 2 (JAK2) inhibitors (Tofacitinib) [8,19]. The unifying mechanism of these medications is immune suppression with inherent increased risk of infection.

Of particular interest is the complex interplay between a patient’s diet, their gut microbial environment and the immune system and how these may be manipulated to better manage their disease, improve efficacy and reduce risks and side effects associated with traditional medical therapies [20]. Patient survey data support this, with a majority of a UK cohort identifying diet as a factor in their IBD [21]. Restrictive diet and avoidance of specific foods in the belief that it could prevent relapse had been utilised by 68% of the cohort. In another survey, up to 89% of IBD patients felt that dietary guidance in IBD was important, yet only 8–16% felt that they received adequate information on the topic from their physicians [22]. Patients with IBD are already at risk of micro- and macro-nutrient deficiencies and this may be exacerbated by unsupervised food avoidance [23]. Whilst it is clear that there has long been the patient perception of the role that diet plays in the initiation and the course of disease, the evidence to support this and how to optimally manipulate dietary intake and provide practical, sound dietary advice has been limited.

Due to chronically poor intake, increased rates of protein turnover and increased gut losses of nutrients during phases of active disease, there may be a net loss of protein. In clinical practice, counselling the IBD patient frequently involves simply broad guidance to improve nutrient intake, including increasing protein intake to 1.2–1.5 g/kg/day during periods of active disease and to avoid foods that may precipitate bowel obstruction. In select adult and paediatric patient groups, exclusive enteral nutrition (EEN) may be used in the short term to manage a flare of disease [4,24].

This review will explore the interactions of diet and the immune system in the context of IBD, focusing on the role of diet in the pathogenesis, particularly the direct and indirect interaction with the immune system. The role of diet as a therapeutic intervention will be discussed to better inform clinical advice and practice in this area. Dietary supplements, pre- and probiotics are an evolving area of therapeutics in the management of IBD and there is a vast amount of literature on this subject. A discussion of these interventions is beyond the scope of this review.

## 3. Diet–Microbial–Immune System Interactions

The microbiome, defined as the collection of microbes and their genes found in their host organ, has become integral to our understanding of health and disease [25]. Dysbiosis, an imbalance in the composition of the gut microbiome, is a hallmark of IBD. This is characterised by loss of microbial diversity, particularly of favourable anaerobic and short-chain fatty acid (SCFA)-producing bacteria such as *Faecalibacterium prausnitzii*, as well as increased numbers of unfavourable adherent and invasive pathogenic species such as adherent-invasive *Escherichia coli* [20,26,27,28,29,30]. Our understanding of the function of the microbiome stems from the field of metabolomics, which is the analysis of the collection of metabolites originating from the activity and function of the gut microbiota; and is continuing to deepen. Recent studies have demonstrated that the differences in function of the gut microbes may be even more pronounced and influential than the changes in the microbial community structure [27,31]. It is postulated that when dysbiosis and altered microbial function combine with the impaired mucosal barrier function found in IBD, mucosal inflammation ensues [32]. Diet is a key determinant of gut microbiota community structure, with changes in the diet leading to altered microbial composition [33]. Diet exerts its influence on the immune system and inflammatory response both through its ability to alter microbial structure and function as well as via interactions with gut mucosal defenses and inflammatory cells directly [34,35]. It follows that the composition of the diet can result in health benefits and, conversely, health risks [36].

To examine the impact that diet may have on the risk of developing IBD, pre-illness diet and subsequent IBD risk has been examined in large cohort studies (Table 1). Limitations in study design include the heterogeneity and inherent inaccuracies of dietary quantification tools, the predominance of studies in females when the incidence rate of IBD has a fairly even sex distribution, the possibility of changes in the diet prior to diagnosis, and the co-existence of different nutritional “insults” in foods (for example, saturated fat and emulsifiers in processed meat products), as well as varied lengths of follow up [32].

Nonetheless, given the many barriers of designing and conducting randomised controlled trials in IBD, if performed rigorously, these observational studies are currently the best evidence available and can provide important clues into the associations between intake of specific dietary components and the risk of developing IBD.

Below we will summarise the proposed mechanisms of diet–microbial–immune system interactions that occur in the pathogenesis of IBD; outlining the current epidemiological evidence for the impact of diet on IBD risk; describing key dietary influences on disease control; and outline the current recommended dietary interventions for active disease.

### 3.1. Breastfeeding

The establishment of the microbiome in childhood is important for its microbial-immune crosstalk that is thought to be involved in the pathobiology of disease later in life [37]. A longitudinal study demonstrated that after an early developmental and transitional phase in gut microbial composition, a period of microbial stability is reached by as early as 31 months [38]. The most significant factor associated with the early microbial structure is whether or not an individual was breastfed, and it also appears that this is a significant risk factor for the development of IBD. A recent meta-analysis found that breastfeeding (BF) was protective against the development of both forms of IBD, reducing the risk of developing CD (odds ratio (OR) 0.71, 95% CI 0.59–0.85) and UC (OR 0.78, 95% CI 0.67–0.91) [39]. This protective effect was accentuated if the duration of breastfeeding was greater than 12 months.

### 3.2. Dietary Fibre

Dietary fibre provides a substrate for the bacteria that inhabit the distal gut, and these bacteria are integral to its handling. Humans produce approximately 17 enzymes to digest fibre. However, these bacteria produce thousands of complementary enzymes to depolymerise and ferment dietary polysaccharides into host absorbable SCFAs. These SCFAs include butyrate, propionate and acetate. Bacteria from the Firmicutes and Bacteroidetes phylum that possess this capability are less abundant in the gut of IBD patients [26]. The IBD-altered microbiota composition results in lower production of anti-inflammatory and immunoregulatory metabolites, in particular butyrate, a lack of which may contribute to increased intestinal inflammation [53].

SCFAs reduce the inflammatory immune response through multiple mechanisms, including reducing the permeability of the intraepithelial barrier; interacting directly with immune cells and indirectly on the production of signaling cytokines and chemokines to collectively decrease proinflammatory mediators and increase immune-tolerant responses. The proposed mechanisms of these impacts are summarised in Table 2 and Table 3 and Figure 1.

Animal models of colitis have shown that dietary fibre intake is protective, particularly if given before the exposure leading to the development of colitis [54]. This may be mediated through the protection from disruption of the gut mucus layer that occurs during active disease [55]. Desai and colleagues demonstrated that fed a low-fibre diet, the gut microbiota in mice resorted to mucus layer glycoproteins as a nutrient source, leading to erosion of the colonic mucus barrier, which serves as a primary defence against enteric pathogens [56]. When rodent pathogen Citrobacter Rodentium was introduced, the mice on the low-fibre diet developed higher rates of colonic inflammation as well as increased areas of mucosal inflammation associated with weight loss and the severity of colitis. This suggests that dietary fibre is important for a healthy mucus barrier, which serves as an important defence mechanism to colonic inflammation. Finally, in vivo butyrate enema treatment has been shown to reduce the number of neutrophils in mucosal crypts and in the surface epithelia of IBD patients with corresponding decrease in disease activity indices [57].

Large cohort studies have explored the relationship between specific components of the diet and the risk of developing IBD. Cohort studies have explored the relationship between dietary fibre intake and IBD. The Nurses’ Health Study prospectively followed 170,776 women for up to 26 years. Intake of fruit-derived fibre in the highest quintile was associated with a 40% risk reduction in the development of CD. Interestingly, there was no impact on UC risk [45]. The Nurses’ Health Study II of 39,511 American women reported a protective effect of fibre intake in CD but not in UC [46]. A smaller case-control study including men and women showed benefit in the subgroup of patient’s with CD who were non-smokers [49]. A recently published systematic review with meta-analysis examined the association of dietary fibre, fruit and vegetable consumption with risk of IBD [58]. There was no significant association between dietary intake of fibre and risk of UC. There was, however, a significant inverse relationship between dietary fibre intake and risk of CD (RR 0.59). When intake of fruit was examined, there was a significant inverse association of fruit intake and the risk of both UC (RR 0.69) and CD (RR 0.47). Similar positive effects were apparent with vegetable intake.

Fibre may also modulate disease activity. A prospective cohort study of 1619 CD and UC patients found that compared with those in the lowest quartile of fibre consumption, participants in the highest quartile were less likely to have a CD flare [59] (Table 2). Participants with CD who reported that they did not avoid high-fibre foods were less likely to have a disease flare than those who avoided food rich in dietary fibre. There was no association between dietary fibre intake and flares in patients with UC. A systematic review found weak evidence that a high fibre intake reduced disease activity in UC. However, this was more marked in fibre supplements rather than dietary changes. Although there was some improvement in symptoms and wellbeing in CD patients with fibre, there was no objective improvement in disease activity [60].

### 3.3. FODMAPs

Diets low in non-digestible short-chain carbohydrates (collectively known as FODMAPs) have been used with success in improving symptoms in patients with functional gut disorders such as Irritable Bowel Syndrome [61]. A randomised, double-blind placebo-controlled study showed higher symptom scores of pain, bloating and urgency in patients with IBD and concurrent IBS challenged with FODMAPs [62]. An improvement in functional symptoms has been demonstrated in cohorts with IBD and concurrent IBS on low FODMAPs diets [63,64]. In IBD patients adhering to a low FODMAP diet, the abundance of select organisms in the stool was decreased, though overall markers of diversity and inflammation were unchanged. One recent study of 60 IBD patients in remission or with mild disease activity demonstrated an improvement in reported symptom scores as well as faecal calprotectin in the subgroup on a low FODMAPs diet compared with a standard diet, though this finding is yet to be replicated [65].

### 3.4. Sugar

Mice fed diets high in sugar have demonstrated proinflammatory changes consistent with IBD [66]. Mice on a 50% sucrose diet have increased gut permeability represented by increased serum lipopolysaccharide (LPS), a bacterial endotoxin and driver of inflammation, decreased microbial diversity and reduced faecal SCFAs (Figure 2). They were also more susceptible to the development of colitis. This has recently been replicated with fructose [67]. It should be noted that this has not yet been reproduced in human trials.

A large nested matched case-control study within the European Perspective Investigation into Cancer (EPIC) study explored the incidence of IBD. Incident IBD was associated with an increased ‘sugar and soft drink’ consumption, with an incident rate of 1.68 [48]. This association was only present when the high sugar and soft drink consumption was concomitant with a low consumption of vegetables. A systematic review found no association between dietary carbohydrate intake and UC risk, except for sucrose consumption, though the pooled relative risk was small at 1.1 per 10 g increment/day of sugar intake [68].

One in ten patients with IBD attribute sugary foods as a cause of worsening of symptoms [21], yet randomised-controlled studies specifically testing the effect of dietary sugar in IBD patients are lacking.

### 3.5. Gluten

Gluten is a factor contributing to intestinal inflammation in TNF-α knockout mice (a mouse model of genetic predisposition to IBD). It is postulated that amylase trypsin inhibitors (ATIs), a family of non-gluten proteins found in gluten containing cereals can regulate the release of inflammatory cytokines, activate Toll-Like Receptors and induce a T-cell response in both coeliac and non-coeliac patients, including those with IBD [69,70].

With up to 28% of IBD patients reporting gluten avoidance in the past, there have been several studies demonstrating subjective improvement in symptoms and decrease frequency of flares on a gluten-free diet in IBD patients without a concurrent diagnosis of coeliac disease [71,72,73]. There was, however, no objective improvement in clinical parameters.

### 3.6. Red Meat

Red meat consumption has been suggested to have a pro-inflammatory effect. This may be due to the cooking method and concurrent presence of saturated fat and its established deleterious effects [74]. Meat-derived protein is subject to fermentation by the gut microflora, which produces branch chain amino acids (BCAAs) as well as potentially toxic substances such as ammonia, amines, hydrogen sulphide, and nitrous compounds. Some of these products of fermentation may potentially damage DNA and promote genetic instability. This effect has been demonstrated in healthy individuals on red meat versus vegetarian diet [75,76]. A colitis mouse model demonstrated that the severity of colitis was increased in mice on a diet high in red meat [77].

A large prospective French cohort (*n* = 67,581) which was followed with second-yearly dietary questionnaires for 15 years demonstrated that a high animal protein intake, specifically meat rather than eggs or dairy, was associated with an increased risk of IBD (HR 1st vs. 3rd tertile 3.31, 95% CI 1.41–7.77) [42]. Furthermore, another large cohort study (*n* = 413,953) reported a significant association between animal protein intake, specifically red meat and UC [51]. There were no significant associations identified with meat or protein consumption and CD. A systematic review of nine studies supported these findings, with red meat conferring a relative risk of IBD of 2.37. On subgroup analysis, this was only significant for UC and not CD. Of note, consumption of poultry was not associated with greater risk of IBD [74].

In UC patients, increased intake of meat (particularly red meat), processed meat, and alcohol was associated with increased risk of flaring [78]. In CD, however, a study which randomised a group of patients to low or high red meat intake for 49 weeks did not find statistically significant difference in rates of disease relapse [79].

### 3.7. Zinc

In rodent models, oral supplementation with zinc has been shown to decrease severity of colitis as well as activity of the myeloperoxidase enzyme (enzyme found in neutrophils and monocytes) when combined with anti-TNF α or as a sole intervention [80,81]. The proposed mechanism was via alterations in TNF receptor expression and flow-on effects to the innate immune response. In mouse and human models of macrophage activity, intracellular zinc enhanced bacterial clearance via autophagy [82]. Data from the Nurses Health Study I and II were combined and food frequency questionnaires on 170,776 women with a follow-up period of 26 years were examined [47]. There was an inverse association between intake of dietary zinc (food sources include meat, fish, cereal and dairy [83] and development of CD, but not UC. Zinc intake has been found to be lower in a Crohn’s disease population compared to healthy controls and lower serum zinc levels have been associated with poorer outcomes such as hospitalization and surgery in both forms of disease [84,85]. Interventional studies examining dietary zinc intake and impact on disease outcomes are lacking.

### 3.8. Vitamin D and Calcium

Osteoporosis is common in patients with IBD. Nutritional deficiencies, including low vitamin D and poor calcium intake are contributing factors, as is chronic inflammation, corticosteroid use and extensive small bowel disease or resection [86].

Vitamin D is synthesised by UV exposure, with smaller amounts being derived from dietary sources such as oily fish and dairy products [87]. Vitamin D preserves and restores epithelial barrier function, in particular via promoting the expression of tight junction proteins [88]. It also inhibits the maturation of dendritic cells and decreases production of co-stimulatory molecules leading to decreased effector T-cell activation [89]. Mice fed a vitamin D-deficient diet had an increased susceptibility to colitis [90]. The immune effects of vitamin D deficiency may be mediated by microbial interactions, with vitamin D and/or vitamin D receptor-deficient mice demonstrating increased bacterial counts in colonic tissue (suggesting bacterial invasion), altered relative abundance of less favourable bacteria including increased Proteobacteria and Bacteroides whilst relative quantities of favourable Firmicutes were decreased [90,91,92].

In a study which examined the Nurses Heath cohort, a validated prediction model of serum vitamin D which included dietary assessment as well as other lifestyle factors was utilised. It found that compared with those with a predicted vitamin D in the lowest quartile, those with predicted levels in the highest quartile had a reduced risk of developing CD but not of UC [44]. This finding was not replicated in the EPIC cohort of 359,728 participants followed for up to 15.7 years. In this study, no significant association between serum vitamin D and IBD risk was found [50].

In a cross-sectional study of 182 CD patients and 62 healthy controls, it was found that low serum levels of vitamin D had an inverse relationship with the Crohn’s Disease Activity Index (CDAI), a score used to quantify patients’ symptoms, with more severe symptoms attracting higher scores [93]. Another retrospective cohort study demonstrated an inverse relationship between serum vitamin D level and scores of disease activity and health-related quality of life in CD and with disease activity scores but not quality of life in UC. [94]. A large retrospective study of 3217 patients with CD and UC found that low levels of vitamin D (<20 ng/mL) were associated with an increased risk of hospitalisation and surgery in both CD and UC. Those who subsequently normalized their level of vitamin D had a reduced risk of surgery compared to those who remained deficient [95]. IBD patients are also at increased risk of osteopenia and osteoporosis, for which low vitamin D is a significant risk factor [96]. It should be noted that studies looking exclusively at dietary vitamin D rather than serum vitamin D (which may be derived largely from UV or supplemental sources) are lacking.

In conjunction with sufficient vitamin D levels, adequate calcium intake is important to ensure optimal bone health. The largest dietary source of calcium is dairy products and IBD patients often avoid this due to its lactose content which is commonly felt to exacerbate symptoms [97]. There are conflicting data on the impact of lactose on the risk of developing IBD, with a recent meta-analysis finding no significant association [98].

In active disease, lactose has been examined as part of the low FODMAPs diet and Specific Carbohydrate Diet (SCD), outlined in more detail below, and these have had favourable results in symptom improvement and, in several small studies, inflammatory activity. As standalone dietary interventions, cow’s milk protein elimination diet and a low-calcium diet in UC and CD respectively have not been shown to be of benefit in inducing or maintaining remission [99,100]. As such, it is important that IBD patients continue to maintain an adequate dietary calcium intake to minimize the risk of bone related complications.

### 3.9. Fat

There are broadly four types of fats: (1) saturated fat found in animal sources such as dairy and meat, food cooked in palm oil and in pastries; (2) monounsaturated fatty acid (MUFA) found in foods such as olive, canola and peanut oils; (3) polyunsaturated fatty acid (PUFA), including omega *n*-3 and *n*-6 PUFA—*n*-3 PUFA is found in canola and linseed oils and in oily fish, and *n*-6 PUFA is found in seed oils such as sunflower—and (4) trans fat found naturally in small amounts in animal sources (e.g., butter) and in larger amounts in hydrogenated food sources [131].

Mouse models have demonstrated that a diet high in saturated fat increased the incidence of colitis. In IL10 knockout mice (a mouse model of genetic predisposition to IBD), a high-saturated-fat diet promoted a pro-inflammatory helper T-cell response and an increased incidence of colitis [102]. The proposed mechanism was that greater taurine (an amino acid found in fatty foods) conjugated with hepatic bile acids, which in turn increases sulphur availability to sulphate-reducing microorganisms such as Bilophila Wadsworthia (of the Deltaproteobacteria species). A combination of a high-fat and high-sugar (HF/HS) diet in mice resulted in increased faecal inflammatory markers [103]. This may be secondary to an altered gut microbiome, with increased proteobacteria in the stool. Mice on these diets were more susceptible to chemically-induced colitis.

High-fat diets have also been shown to reduce the level of secretory Immunoglobulin A (IgA) coating the gut microbiota [104].

A large prospective cohort study demonstrated a protective effect of higher consumption of the *n*-3 PUFA docosahexaenoic acid (DHA) and the development of UC [43]. However, a meta-analysis did not find a significant association between total fat consumption, or its subtypes, and risk of developing UC [132].

A study compared changing dietary intake and incidence of CD in a homogenous Japanese population two decades apart. They found that the incidence of CD was significantly increasing in this population, who had minimal migration and therefore stable genetics. As such, confounding variables were minimised. Total fat (r = 0.919), animal fat (r = 0.88) and the ratio of *n*-6/*n*-3 fatty acid intake (r = 0.792) were all significantly correlated with CD risk [40]. A case-control study in a paediatric population found that the consumption of omega 3 fatty acids as well as a higher ratio of *n*-3/*n*-6 fatty acids was associated with reduced CD risk [41].

In the previously described Nurses Health Cohort study, there was a trend towards a lower risk of UC in those with a greater *n*-3 PUFA intake and a trend towards an increased risk of UC in those with a high intake of transfats, neither of which reached statistical significance [133]. There was no association between fat intake and CD risk.

Dietary fat has been demonstrated to have an impact on disease activity. A prospective study of 412 patients with UC in clinical remission found an association between high dietary intake of myristic acid, a saturated fatty acid found in coconut and palm oils and dairy products and flare of disease [109]. The ratio of *n*-6/*n*-3 PUFA intake also appears to be important. *n*-6 PUFAs are precursors for proinflammatory cytokines and are ligands for nuclear receptors that have a critical role in inflammatory signaling pathways. *n*-3 PUFA, on the other hand, are suggested to be anti-inflammatory, being precursors to anti-inflammatory cytokines and suppressing inflammatory T-cell responses [105,106,107]. Uchiyama and colleagues examined the influence of the advice to increase dietary intake of *n*-3 PUFA and reduce *n*-6 PUFA in a cohort of IBD patients and demonstrated that remission at 12 month follow up was associated with a higher ratio of *n*-3/*n*-6 PUFA in the red blood cell membrane than those who had relapsed [108].

### 3.10. Emulsifiers and Nanoparticles

Emulsifiers are detergent-like molecules used in processed foods that are added to improve texture and quality [134]. However, they may interfere with gut barrier function [135]. In vitro studies have shown significantly increased translocation of *E. coli* through mucosal cells with the addition of emulsifying agent compared with vegetable-based fibres [111]. In mouse models, the addition of emulsifiers that are commonly used in human food have been associated with decreased mucus thickness, narrowed distance of bacteria to mucosal layer, altered compositions of bacteria, including decreased Bacteroidetes, increased pro-inflammatory cytokines and a higher incidence of and more extensive colitis in genetically predisposed mice [110].

In an ex vivo human intestinal model, it was demonstrated that the addition of the emulsifiers polysorbate 80 (P80) and carboxymethylcellulose (CMC) did not have an impact on pathogen-free mice, suggesting that the mechanism of influence is mediated through microbial interactions [136]. When P80 (commonly found in ice-cream) was added to a human microbial model, there was a 50% decrease in species diversity, an increase in microbial genes that coded for flagellin expression, a marker of bacterial adhesion and invasion that can directly activate an inflammatory response [137]. In the next phase of the experiment, this model was injected into a mouse host; increased IL6 (a pro-inflammatory cytokine) expression was observed. Thus, it appears that emulsifiers contribute to inflammation via a microbial dependent pathway. Carrageenan is another food thickener and stabiliser used in flavoured milk and yogurt and it was found in a review of forty-five studies examining animal models to increase the incidence of gastrointestinal ulceration and neoplasia [138].

Maltodextran is a modified starch molecule that is commonly added to processed foods to optimise texture. It has been found in mouse models that its administration increased *E. Coli* biofilm formation, which has the potential to detrimentally affect the gut mucosal barrier [139]. In ileal sampling of CD patients, they were found to have increased levels and prevalence of the malX gene (coding the enzyme that metabolises Maltodextran) when compared with controls [140]. This strengthens the theory that this substance increases susceptibility to invasive bacterial species associated with IBD as part of the mechanism of increased intestinal inflammation [134].

Nanoparticles (including titanium dioxide and aluminum silicates) are used as food additives to preserve food and in the cosmetics industry, and are also found in some pharmaceutical formulations. In animal models, orally administered nanoparticles have been shown to accumulate in intestinal epithelium, activate the NOD-, LRR- and pyrin domain containing protein 3 (NLRP3) inflammasome and increase intestinal inflammation. The authors found that patients with active UC had significantly higher blood levels of titanium than patients in remission as well as in healthy individuals [112].

Roberts and colleagues demonstrated strong associations between geographic locations of increased CD incidence with parallel increased emulsifier consumption in Japan, Europe, Canada and the US [141].

### 3.11. Mediterranean Style Diet

A recent prospective cohort study examined the relationship between consumption of a Mediterranean diet (high in fruit, vegetables, PUFA and protein predominately from fish, legumes and nuts) and the risk of developing CD in a prospective cohort study of 83,147 Swedish adults [52]. They found a dose–effect relationship, with those most adherent to the Mediterranean diet having a lower risk of developing CD. A study examined the dietary intake of 39,511 young women during high school and subsequent development of IBD over 19 year follow up. They found that compared to women who consumed <10 g/day of fish during high school, those with an intake of ≥30 g/day had a 57% lower risk of CD [46]. Fish intake did not appear to modulate UC risk.

In summary, while epidemiological nutritional studies are limited by many confounders, some relationships were strong and consistent across studies. Breast milk, fruit and vegetable intake and an increased ratio of *n*-3/*n*-6 fatty acids in the diet appear protective, whereas sugar containing soft drink intake is associated with a greater risk of both forms of IBD. Red meat appears to only be related to risk of UC, and an increased emulsifier and animal fat intake appears to increase the risk of CD. The strongest evidence for risk reduction in the development of CD is with fish, particularly as part of a Mediterranean diet as well as adequate zinc and vitamin D intake, whilst DHA intake may reduce UC risk.

Disease activity may also be modulated by diet, with evidence that dietary fibre, a high dietary *n*-3/*n*-6 PUFA ratio may decrease flare risk in both forms of IBD. Factors found to increase flare risk in UC include myristic acid and red meat. Zinc and vitamin D levels have been inversely associated with levels of disease activity and poorer health outcomes in both forms of IBD. Diets low in gluten and FODMAPs have been shown to improve residual symptoms in those with IBD in remission, specifically in those with concomitant IBS.

## 4. Diet as Prescribed Therapy for Active Disease in IBD

There are several prescriptive dietary approaches that have been studied for use in routine clinical practice to manage IBD flares (Table 4). Due to the restrictive nature of these dietary regimens, they are largely reserved for the most unwell patients or those wanting to avoid the use of steroid-based medications, particularly children in whom steroids can impact upon growth trajectory [142].

### 4.1. Exclusive Enteral Nutrition (EEN)

Dietary food components may contribute to the inflammatory processes during a flare of IBD. EEN is an elemental diet, which involves the provision of 100% of a person’s nutritional requirement as a liquid nutrition formula delivered enterally (either orally or via nasogastric tube) over a period of 6–8 weeks [149]. The mechanism of its efficacy is still poorly understood. However, there are changes in the gut microbiome and metabolome [150]. EEN significantly decreases the gut bacterial diversity below pre-treatment levels. One theory is that the bowel rest induced by EEN facilitates mucosal healing by limiting the activity of pathogenic microbes [149]. EEN has been found to be particularly beneficial in the paediatric setting and in CD rather than UC. A randomised controlled trial that looked at both clinical remission and mucosal healing at week 10 in a paediatric CD population comparing EEN and oral corticosteroids showed equivalent rates of clinical remission and superior rates of mucosal healing in the EEN group [143]. Studies looking at the utility of EEN in the adult IBD population have yielded less dramatic benefits. Meta-analyses on the topic have suggested that there is weak evidence of the benefit of EEN for the induction and maintenance of IBD remission compared to placebo in adult patients with CD, but not compared to corticosteroids [151,152]. One of the main barriers to efficacy is patient adherence to EEN therapy [144].

There is marked heterogeneity in the macronutrient and additive composition of EEN formulas. An informative study was performed which compared the nutritional and additive content of 61 formulas with published efficacy in the management of active CD with equivalent rates of remission [153]. Interestingly, many dietary components that have been implicated in triggering CD, such as high *n*-6/*n*-3 PUFA ratios (derived from vegetable oils), refined sugar and additives such as emulsifiers, maltodextrin and carrageenan, were contained in these formulas despite their efficacy in the treatment of active CD. This may suggest that despite epidemiological, pre-clinical and select clinical data suggesting benefits of excluding these dietary components, their impact in the context of active disease cannot be assumed to be significant without further well-designed prospective clinical studies. Of note, all formulas were lactose free and gluten free and 82% lacked fibre, suggesting that these macronutrients may be aspects of dietary management that are important considerations in the context of active disease.

### 4.2. Partial Enteral Nutrition (PEN)

It was hoped that the delivery of partial enteral nutrition (PEN: 50% enteral nutrition and 50% unrestricted diet) may improve compliance. When PEN was compared with EEN, the EEN regimen was far superior [145]. In an attempt to optimise the PEN approach, Levine and colleagues designed a 12 week study comparing standard EEN and a specific CD exclusion diet (CDEC) plus PEN in a paediatric population [146]. The CDED consisted of foods that were least likely to have adverse effects on the microbiome and intestinal barrier. Allowed foods included chicken, bananas and potatoes, dairy and wheat; food rich in emulsifiers and artificial sweeteners was avoided. The patient acceptability rate was higher with the CDED as opposed to EEN and the clinical response rates were not statistically different between the two diet regimens. This shows promise that a less restrictive dietary approach can be used and achieve non-inferior remission rates to EEN therapy.

### 4.3. Real Food Diet

The CD-TREAT study evaluated the effects of an individualised food-based diet, which aimed to re-create the benefits of an EEN diet by exclusion of specific dietary components, such as gluten, lactose and alcohol. It was found to have a similar impact to EEN on the gut microbiome, inflammation and clinical response in a rat model as well as in healthy adults and children with CD. This diet was easier for patients to comply with and more acceptable than EEN [147].

### 4.4. Specific Carbohydrate Diet

The Specific Carbohydrate Diet (SCD) is based on an exclusion of complex carbohydrates, sugar, most dairy and processed foods that are believed to be poorly absorbed (or “pro-inflammatory”) [107]. Allowed foods include most nuts and vegetables, fermented yogurt and fresh, unprocessed meat, poultry and fish. Studies to date have been retrospective, without objective markers of IBD clinical activity or small case series [154,155].

In a small study, nine paediatric CD patients with mildly active disease at recruitment were instructed on the SCD, received monthly sessions with a dietician to optimise adherence and were followed up for 52 weeks [101]. The authors found a significant reduction in clinical indices of severity as well as increased mucosal healing rates as assessed by small bowel capsule endoscopy (a swallowed pill size camera that enables visual assessment of the otherwise difficult to access small bowel). Another pilot study of the SCD in paediatric patients with UC and CD demonstrated that adherence led to reduced serum and stool inflammatory markers associated with disease activity [148]. Significant changes in gut microbial composition were noted 12 weeks into this dietary change in some individuals. These changes included increased diversity and decreases in invasive, adherent species such as *E. coli*. This raises the possibility of using a patient’s microbiome as a monitoring tool in instituting dietary therapy.

### 4.5. Nanoparticles

A pilot study of 20 CD patients with active disease were randomised to two groups. The first group was given standard care plus dietary advice to avoid food containing nanoparticles (such as packed ready meals, processed meat, flavoured milk and processed cheese) and a second control group received standard care alone. They found a significant improvement in the Crohn’s Disease Activity Index (CDAI) at 4 months in the intervention group [113]. A larger multicentre study, however, did not find a significant benefit of diet restricted in nanoparticles at 4 or 12 months [99].

## 5. Future Directions of Diet and IBD—Personalised Nutrition

It is increasingly clear that in clinical a one-size-fits-all approach is not appropriate. With improved technology we will be able to tailor an individual’s treatment based on specific factors, termed precision medicine. In IBD, factors such as diet, genetics, microbiome, clinical parameters and biomarkers will need to be considered in order to design the most effective treatment strategy [156,157,158]. Specific to diet, this approach has been demonstrated in a seminal paper, which explored diet in a generally healthy population. Post-prandial glucose responses (PPGR) were accurately predicted using a machine learning algorithm that combined individual participant features including diet, physical activity and gut microbiota [159]. Personalised dietary interventions led to improved PPGRs with concomitant shifts in the microbiome in a way that has been previously associated with lowered metabolic risk. These findings have been replicated in a second cohort [160]. This suggests that PPGR patterns may give a valid readout of the makeup of an individual, including the gut microbiome. A multifactorial approach using machine learning such as this could lead to specific dietary recommendations that would favourably change a patient’s microbiota, an attractive prospect.

In summary, it is clear that the use of diet in the prevention and treatment of IBD is a desirable treatment strategy. It is therefore important that clinicians have a good knowledge of the specific components of diet that exert both pro-inflammatory and anti-inflammatory effects in the context of IBD. These effects are mediated through influences on the mucosal barrier function, direct immune system interactions and the gastrointestinal microbiome.

There is epidemiological evidence in humans from large population cohort studies that diet exerts a strong influence on the development and course of IBD. Most influential are the protective effect of breastfeeding, fruit-derived fibre, and the Mediterranean diet. In contrast, high sugar, red meat and processed food intake likely increase IBD risk. Low zinc and vitamin D levels have been associated with CD and but not UC.

Most evidence based medical treatments for IBD are directed at suppressing the immune response and carry risks of significant side effects. The current evidence has demonstrated several dietary factors that are likely to be protective against flares, such as fibre, zinc and vitamin D in both forms of IBD and a high dietary *n*-3/*n*-6 PUFA ratio in UC. Other factors found to increase flare risk are myristic acid containing foods and red meat in UC and sugar in CD.

It is likely that diet as a prevention and therapeutic intervention will find its place in a personalised patient treatment approach that takes into account individual patient factors to achieve optimal disease control and minimise unwanted side effects.

## Figures and Tables

**Figure 1 nutrients-13-00135-f001:**
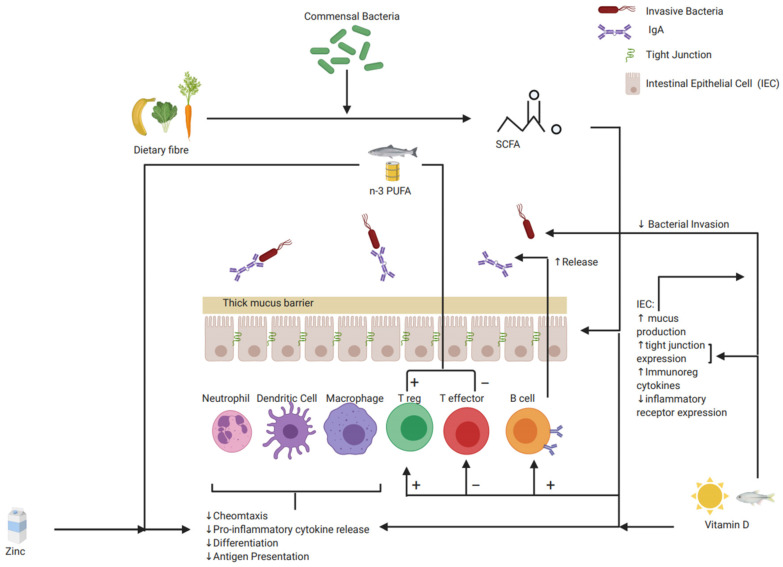
Schematic representation of proposed mechanism of favourable diet–gut immune interaction that may protect from the development of IBD and improve disease activity (Created with BioRender). Abbreviations: polyunsaturated fatty acid (PUFA), short-chain fatty acid (SCFA), and immunoglobulin (Ig).

**Figure 2 nutrients-13-00135-f002:**
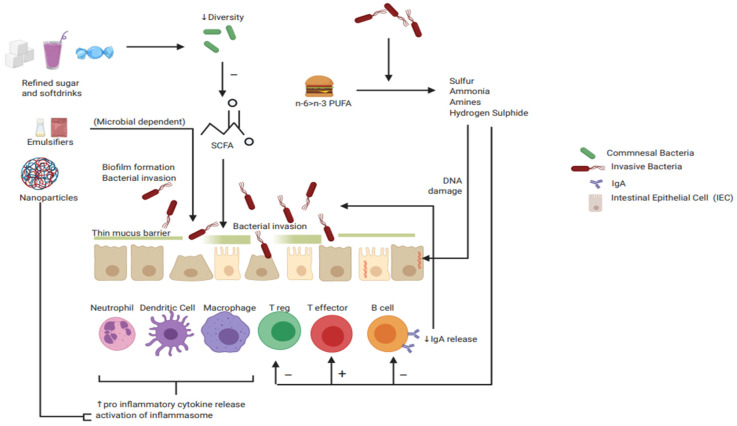
Schematic representation of proposed mechanisms of pathological diet-immune interactions that may contribute to IBD risk and disease activity. (Created with BioRender). Abbreviations: polyunsaturated fatty acid (PUFA), short-chain fatty acid (SCFA), Ig (Immunoglobulin).

**Table 1 nutrients-13-00135-t001:** Key epidemiological studies of the impact of diet on IBD risk.

Study	Year	Design	Sample Size	Follow up (Years)	IBD Type	Exposure	Impact on Risk
Shoda [40]	1996	Epidemiologic	68,000	12	CD	Correlations between dietary intake and CD risk	Total fat r 0.919 (*p* < 0.01) * Animal fat r 0.88 (*p* < 0.01) * Ratio *n*-6/*n*-3 PUFA 0.792 (*p* < 0.01) *
Amre [41]	2007	Case control	202 controls 120 CD	Dietary consumption 1 year prior to diagnosis	CD	*n*-3 PUFA, highest vs. lowest quartile; Ratio *n*-3 PUFA/*n*-6 PUFA, highest vs. lowest quartile	OR 0.44 (95% CI 0.19–1.0) OR 0.32 (95% CI 0.14–0.71) *
Jantchou [42]	2010	Prospective cohort	67,581 participants Incident IBD cases: 77	10.4	CD and UC combined	Animal protein, 3rd vs. 1st tertile	HR 3.03 (1.45–6.34) *
John [43]	2010	Prospective cohort	25,639 participants Incident UC cases: 22	Mean 4.2 (1.8–4.2)	UC	Intake of DHA in the highest tertile	UC OR 0.43 (95% CI 0.22–0.86) *
Anathkrishnan [44]	2012	Prospective cohort	72,719 Incident cases: 122 CD 123 UC	22	CD and UC	Validated prediction score of serum vitamin D level	CD HR (95% CI 0.3–0.99) UC 0.65 (95% CI 0.34–1.25)
Ananthakrishnan [45]	2014	Prospective cohort	170,776 Incident cases: CD 269 UC 338	26	CD and UC	*n*-3 PUFA intake in highest quintile; transfat intake in highest quintile	UC HR 0.72 (95% CI 0.51–1.01) UC HR 1.34 (95% CI 0.94–1.92)
Ananthakrishnan [45]	2014	Prospective cohort	170,776 Incident cases: CD 269 UC 338	26	CD and UC	Fibre intake, highest quintile	CD HR 0.59 (95% CI 0.39–0.9) * UC HR 0.82 (95% CI 0.58–1.17)
Ananthakrishnan [46]	2015	Prospective cohort	39,511 Incident cases: CD 70 UC 103	26	CD and UC	Fibre intake, highest quartile	CD HR 0.47 (95% CI 0.23–0.98) * UC: not significant
Anathkrishnan [47]	2015	Prospective cohort	170 776 Incident cases CD 269 UC 338	26	CD and UC	Dietary zinc intake, highest quartile vs. lowest quartile	CD HR 0.63 (95% CI 0.43–0.93) * UC HR 0.96 (95% CI 0.68–1.34)
Ananthakrishnan [46]	2015	Prospective cohort	116,686 Incident cases CD 70 UC 103	19	CD and UC	Fish intake, highest vs. lowest quartile	CD HR 0.43 (95% CI 0.21–0.90) * UC HR 0.99 (95% CI 0.55–1.77)
Racine [48]	2016	Prospective cohort	366,351 participants Incident cases CD 117 UC 256	18	CD and UC	High sugar and soft drinks intake, 5th vs. 1st quintile (>2 years post-dietary assessment)	UC IRR 1.68 (1.00–2.82) * No impact on CD risk
Anderson [49]	2018	Case control (from prospective cohort)	401,326 CD controls, CD cases 104 UC controls 884, UC cases 221	19	CD and UC	Dietary fibre intake, quartile 4 vs. 1	CD non-smokers OR 0.5 (95% CI 0.29–0.86) * CD OR 0.83 (0.38–1.81) UC OR 1.22 (0.71–2.08)
Opstelten [50]	2018	Prospective cohort	359,728 Incident cases: CD 72 UC 169	Up to 15.7 years	CD and UC	Serum vitamin D at baseline, highest vs. lowest quartile	CD OR 0.69 (95% CI 0.29–1.6) UC OR 1.22 (95% CI 0.67–2.2)
Dong [51]	2020	Prospective cohort	413,953 participants CD 177 UC 595	16	CD and UC	Animal protein intake increase/10 g/day; red meat, 4th vs. 1st quartile	HR 1.1 (95% CI 1.004–1.21) * HR 1.41 (95% CI 1.03–1.92) * (Only significant for UC)
Khalili [52]	2020	Prospective cohort	83,147 Incident cases: CD 164 UC 395	17	CD and UC	Adherence to Mediterranean diet (highest vs. lowest Med score)	CD HR 0.42 (95% CI 0.22–0.80) * UC HR 1.08 (95% CI 0.74–1.58)

Abbreviations: inflammatory bowel disease (IBD), ulcerative colitis (UC), Crohn’s disease (CD), confidence interval (CI), odds ratio (OR), hazard ratio (HR), incident rate ratio (IRR), polyunsaturated fatty acid (PUFA), and docosahexaenoic ACID (DHA). (* = statistically significant).

**Table 2 nutrients-13-00135-t002:** Dietary components, proposed mechanism of influence and impact on disease control.

Dietary Component	Proposed Mechanism	Impact on Disease Control	
		Crohn’s Disease	Ulcerative Colitis
Fibre	Fermented to SCFA by colonic bacteria. Downstream effects include - Strengthen mucosal barrier function including thickened mucus layer and promote IgA production - Decrease inflammatory cytokines production - Suppress development, maturation and function of inflammatory cells [34,35]	Reduces risk of flares [59]	Improves disease activity [60]
FODMAPs	Fermented by commensal bacteria in distal gut to produce gas and distension [62]	Exclusion improves symptoms [63,65]	Exclusion improves symptoms [63,65]
Wheat/Gluten	ATI proteins stimulate release of inflammatory cytokines, activate Toll-Like Receptors and induce a T-Cell response [69,70]	Improved symptoms, especially in stricturing CD [71,72,73]	Improved symptoms [71,72,73]
Sugar	Increased gut permeability Decreased microbial diversity Increased pro-inflammatory cytokines [66,67]	Improve clinical indices and mucosal healing in a small paediatric study [101]	Reported by patients but no trials to support [21]
Fat	Saturated fat>increased taurine conjugation of hepatic bile acids>increases sulphate reducing microorganisms>promote pro-inflammatory T-cell response [102,103,104] *n*-3 PUFA anti-inflammatory vs. *n*-6 PUFA pro-inflammatory [105,106,107]	Higher *n*-3 PUFA: *n*-6 PUFA ratio improves remission rates [108]	High dietary fat intake [78], in particular myristic acid (found in palm oil, coconut oil and dairy fat) associated with flares [109]
Emulsifiers	Erode mucus barrier [110] Increase translocation of invasive *E. Coli* [111] Increase pro-inflammatory cytokines [110]	No studies in human (apply everywhere)	No studies in humans
Nanoparticles	Activate inflammasome [112]	Small pilot study showed improved remission rates [113], not replicated in larger study [99]	No studies in humans
Meat	Fermentation produces by-products that promote DNA instability [75,76] Also high in fat (see above)	No impact demonstrated [79]	Red meat increased risk of disease flares [78]
Zinc	Modulates TNF expression which decreases myeloperoxidase enzyme activity [80,81,82]	Low zinc levels associated with increased hospitlaisation and surgery [85]	Low zinc levels associated with increased hospitlaisation and surgery [85]
Vitamin D	Strengthens epithelial barrier and tight junction protein expression [88] Decreases maturation and function of Dendritic cells with subsequent decreased T-cell activation [89]	Inverse relationship between serum vitamin D and CDAI and HBI as well as poorer health-related QOL scores [93,94]	Inverse relationship between serum vitamin D and UCDI [94]
Calcium		Low-calcium diet did not improve disease control [99]	Cow’s milk protein exclusion did not improve disease control [100]

Abbreviations: polyunsaturated fatty acid (PUFA); short-chain fatty acid (SCFA); Crohn’s disease (CD); amylase trypsin inhibitor (ATI); Fermentable, Oligosaccharides, Disaccharides, Monosaccharides, and Polyols (FODMAPs); Crohn’s Disease Activity Index (CDAI); Harvey–Bradshaw Index (HBI); quality of life (QOL); Ulcerative Colitis Disease Activity Index (UCDI); tumour necrosis factor (TNF).

**Table 3 nutrients-13-00135-t003:** Immune interactions of short-chain fatty acids (SCFAs).

Immune Cells	Intestinal Epithelial Cells: Barrier	Pro Inflammatory Mediators	Anti-Inflammatory Mediators
- SCFA are ligands of G-protein-coupled receptors> the production of pro inflammatory mediators> decrease neutrophil chemotaxis [114,115]. - Dendritic cell and macrophage differentiation, and maturation, co stimulation & decrease production of pro inflammatory cytokines [116]. Results in tolerance to commensals rather than inflammation. - Act on T cells to Inhibit antigen presentation inhibit differentiation & proliferation promote apoptosis of CD4+ and CD8+ effector T cells [117] regulatory T cells [118,119]	IEC production of mucin and antimicrobial peptides [120,121,122] Modulate the expression of tight junction proteins [123] which strengthens intestinal barrier function. IEC expression of anti-inflammatory IL-18 and TNF-β [124] IEC expression of pro inflammatory receptors, cytokines and chemokines including Toll Like Receptor 4 (TLR4) [125,126], IL-8 [127], CXCL5 and CXCL11 [128]	- recruitment of monocytes and neutrophils to the gut by reducing the release of pro inflammatory chemokines e.g., monocyte chemo attractant protein-1 (MCP-1) & Vascular cell adhesion molecule-1 (VCAM1) [129] - surface expression of chemoattractant receptors C5aR and CXCR2 [53] - the release of pro inflammatory cytokines e.g., TNF α and Nitric Oxide by neutrophils [130] - Inhibit macrophage production of pro inflammatory mediators TNF alpha and IL-6 [53]	- Modulate IgA transcytosis to gut lumen by causing agglutination and anchorage in the lumen to prevent bacterial contact with epithelium [53]. - Promote production of IL-10 and IL-5 by T cells [53].

Abbreviations: short-chain fatty acids (SCFAs), intestinal epithelial cell (IEC), immunoglobulin A (IgA), interleukin (IL), tumour necrosis factor (TNF), chemokine receptor (CXC), vascular cell adhesive molecule (VCAM), intestinal epithelial cell (IEC), and T-helper cells (CD4+).

**Table 4 nutrients-13-00135-t004:** Summary of dietary interventions in Crohn’s disease.

Diet	Description	Evidence
Exclusive Enteral Nutrition	100% of nutritional requirements delivered via liquid formula orally or via nasogastric tube	Equivalent to corticosteroids in achieving remission in paediatric population [143] Adherence is challenging in adults [144]
Partial Enteral Nutrition	50% Enteral nutrition plus 50% unrestricted diet	Inferior to EEN [145]
	50% Enteral nutrition plus 50% select whole foods	Equivalent to EEN in response rates Improved patient acceptability compared to EEN [146]
Ordinary Food Diet (CD-TREAT)	Food based diet with exclusion of specific components in common with EEN such as gluten, lactose and alcohol	Reduced inflammatory markers and disease activity at week 12 and better tolerated than EEN (though *n* = 5) [147]
Specific Carbohydrate Diet	Excludes grains, sugars, processed food and most dairy	Reduced inflammation at 12 weeks [148]
Reduced microparticles diet	Dietary advice to avoid food containing micro particles	Pilot study of 20 patients demonstrated improvement in CDAI at 4 months [113], not replicated in larger multicentre 82 patient cohort [99]

Abbreviations: exclusive enteral nutrition (EEN) and Crohn’s Disease Activity Index (CDAI).

## Data Availability

Not applicable.

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
