# Peer review of "The Role of Diet in the Pathogenesis and Management of Inflammatory Bowel Disease: A Review"

_nutrients, 2020, doi:10.3390/nu13010135_

Round 1
Reviewer 1 Report
Comments to the Authors of manuscript number: nutrients-1051141 entitled “The role of diet in the pathogenesis and management of Inflammatory Bowel Disease: A review”.
Inflammatory bowel disease is very common problem, which can last for few decades of life as a chronic disease. Given its frequent occurrence in children, it can have an incredibly large impact on a teenager's ability to lead a normal life, on the other hand it is present in many elders. IBD is manifested in many cases by parenteral symptoms such as anemia, disorders of the musculoskeletal system, ocular, hepatic and dermatological complications. Problems in treating IBD are from its relationship with our immune system.
The review is written very well, includes many different aspects of the diet in IBD, supported by published papers. However, in my feeling there is a lack some points.
- The most common causes of nutritional disorders in IBD. They include:
- insufficient oral nutrition (e.g. due to anorexia, anorexia, side effects of drugs) • increased loss of nutrients (malabsorption and / or digestion, loss through fistulas, etc.) • increased need for nutrients (exacerbation of disease, infections, fever).
If Authors wrote in details about the diet, I think that it is good to mention about disorders of nutritional status caused by IBD.
- the role of vitamin D in the diet should be added. There are studies which showed a beneficial effect of vitamin D on the reduction of the incidence of UC, but not chronic colitis
- the same with zinc. consumption of> 16 mg of zinc per day, especially in the natural form, not in the form of a dietary supplement, reduced the risk of developing chronic colitis
- Patients with IBD are at high risk of losing lean body mass, the main causes of which are: decreased food intake, increased protein turnover, loss of nutrients from the gastrointestinal tract, and drug side effects. Thus, it should be mentioned that patients with IBD in remission have a demand similar to that of the healthy population (about 1 g / kg bw / day), while in the event of an exacerbation of the disease, the protein supply should be increased to 1.2–1.5 g / kg bw / day.
- It is worth to add about the importance of Ca in the diet, because both in adults and children suffering from IBD, as well as in patients treated with steroids, calcium and vitamin D levels should be monitored, and their deficiencies should be supplemented. Osteopenia and osteoporosis should be managed according to the general standards of medical science.
- The use of probiotics can be considered. (E. coli Nissle 1917 or VSL#3).
Miele E., Pascarella F., Giannetti E. at all.: Effect of a probiotic preparation (VSL#3) on induction and maintenance of remission in children with ulcerative colitis. Am. J. Gastroenterol., 2009; 104: 437–443
Ruemmele F.M., Veres G., Kolho K.L. at all.: Consensus guidelines of ECCO/ESPGHAN on the medical management of pediatric Crohn’s disease. J. Crohns Colitis, 2014; 8: 1179–1207
Author Response
Response to Reviewer 1 Comments
Comments to the Authors of manuscript number: nutrients-1051141 entitled “The role of diet in the pathogenesis and management of Inflammatory Bowel Disease: A review”.
Inflammatory bowel disease is very common problem, which can last for few decades of life as a chronic disease. Given its frequent occurrence in children, it can have an incredibly large impact on a teenager's ability to lead a normal life, on the other hand it is present in many elders. IBD is manifested in many cases by parenteral symptoms such as anemia, disorders of the musculoskeletal system, ocular, hepatic and dermatological complications. Problems in treating IBD are from its relationship with our immune system.
The review is written very well, includes many different aspects of the diet in IBD, supported by published papers. However, in my feeling there is a lack some points.
- The most common causes of nutritional disorders in IBD. They include:
- insufficient oral nutrition (e.g. due to anorexia, anorexia, side effects of drugs) • increased loss of nutrients (malabsorption and / or digestion, loss through fistulas, etc.) • increased need for nutrients (exacerbation of disease, infections, fever).
Response 1: Thank-you for your comment, see revised version of manuscript. (See lines 41-44)
If Authors wrote in details about the diet, I think that it is good to mention about disorders of nutritional status caused by IBD.
- the role of vitamin D in the diet should be added. There are studies which showed a beneficial effect of vitamin D on the reduction of the incidence of UC, but not chronic colitis
Response 2a: Agree this should be included, see revised version of manuscript (See lines 294-330)
the same with zinc. consumption of> 16 mg of zinc per day, especially in the natural form, not in the form of a dietary supplement, reduced the risk of developing chronic colitis
Response 2 b: Agree this should be included, see revised version of manuscript (See lines 279-292)
- Patients with IBD are at high risk of losing lean body mass, the main causes of which are: decreased food intake, increased protein turnover, loss of nutrients from the gastrointestinal tract, and drug side effects. Thus, it should be mentioned that patients with IBD in remission have a demand similar to that of the healthy population (about 1 g / kg bw / day), while in the event of an exacerbation of the disease, the protein supply should be increased to 1.2–1.5 g / kg bw / day.
Response 3: Agree this should be included, see revised version of manuscript (See lines 94-99)
- It is worth to add about the importance of Ca in the diet, because both in adults and children suffering from IBD, as well as in patients treated with steroids, calcium and vitamin D levels should be monitored, and their deficiencies should be supplemented. Osteopenia and osteoporosis should be managed according to the general standards of medical science.
Response 4: See updated manuscript, (lines 332-344)
- The use of probiotics can be considered. (E. coli Nissle 1917 or VSL#3).
Response 5: Dietary supplements, pre and pro biotics are an evolving area of therapeutics in the management of and there is a vast amount of literature on this subject. A discussion of these interventions is beyond the scope of this review. (See updated manuscript 106-108)
Miele E., Pascarella F., Giannetti E. at all.: Effect of a probiotic preparation (VSL#3) on induction and maintenance of remission in children with ulcerative colitis. Am. J. Gastroenterol., 2009; 104: 437–443
Ruemmele F.M., Veres G., Kolho K.L. at all.: Consensus guidelines of ECCO/ESPGHAN on the medical management of pediatric Crohn’s disease. J. Crohns Colitis, 2014; 8: 1179–1207
Reviewer 2 Report
Current manuscript entitled "The role of diet in the pathogenesis and management of IBD: a review" is a very interesting and well written manuscript about the effects of various dietary parameters and patterns on IBD risk and treatment.
However, please consider adding the following comments:
- line 70 please correct the statement "is of CD''
Table 1: please list studies in table according to publication date ( for example 2010, 2011, 2012, 2013..........)
line 192 - please consider the following studies about fodmap
Cox et al. Gastroenterology. 2020 Jan;158(1):176-188
line 103 - please consider writing few things about the study by Logan et al. Al Pharm Ther 2020 May;51(10):935-947.
References - please correct references according to MDPI directions
Author Response
Response to Reviewer 2 Comments
Current manuscript entitled "The role of diet in the pathogenesis and management of IBD: a review" is a very interesting and well written manuscript about the effects of various dietary parameters and patterns on IBD risk and treatment.
However, please consider adding the following comments:
Comment1: line 70 please correct the statement "is of CD''
Response 1: Thank-you for your comment, see updated manuscript (Line 76)
Comment 2: Table 1: please list studies in table according to publication date ( for example 2010, 2011, 2012, 2013..........)
Response 2: Agree this is an improved ordering system, see Table1 .
Comment 3: line 192 - please consider the following studies about fodmap Cox et al. Gastroenterology. 2020 Jan;158(1):176-188
Response 3: This is an up-to-date and informative article. See updated manuscript (lines 218-220) and ref 64
Comment 4: line 103 - please consider writing few things about the study by Logan et al. Al Pharm Ther 2020 May;51(10):935-947.
Response 4: Thank-you for your comment and suggestion to add this novel study. Please see updated manuscript (pages 137-148)
Comment 5: References - please correct references according to MDPI directions
Response 5: Thank-you for pointing this out, I have appended font and format,